

# Identification and charactering of APX genes provide new insights in abiotic stresses response in *Brassica napus*

Jiao Pan[1,2], Lei Zhang[1,2], Min Chen[1,2], Yuxuan Ruan[1,2], Peifang Li[1,2], Zhihui Guo[1,2], Boyu Liu[1,2], Ying Ruan[1,2], Mu Xiao[1,2] and Yong Huang[1,2]

[1] College of Bioscience and Biotechnology, Hunan Agricultural University, Changsha, Hunan, China
[2] Key Laboratory of Crop Epigenetic Regulation and Development in Hunan Province, Changsha, Hunan, China

## ABSTRACT

Ascorbate peroxidase (APX) plays an important role in scavenging $H_2O_2$ and balancing ROS content in plant cells, which is of great significance for the growth and development of life and resistance to external stress. However, up to now, APXs in *Brassica napus* (*B. napus*) have not been systematically characterized. In this study, a total of 26 *BnaAPX* genes were identified, which were distributed on 13 chromosomes and divided into five phylogenetic branches. Gene structure analysis showed that they had a wide varied number of exons while BnaAPXs proteins contained more similar motifs in the same phylogenetic branches. qRT-PCR analysis of 26 *BnaAPX* gene expression patterns showed that three putative cytosol *BnaAPX* genes *BnaAPX1*, *BnaAPX2*, *BnaAPX9*, two putatice microsomal genes *BnaAPX18* and *BnaAPX25* were up-regulated rapidly and robustly under high salt, water shortage and high temperature stresses. In addition, the above three abiotic stresses led to a significant increase in APX activity. The results provide basic and comprehensive information for further functional characterization of APX gene family in *B. napus*.

# INTRODUCTION

Reactive oxygen species (ROS), which includes superoxide anion radicals ($O_2^{.-}$), hydrogen peroxide ($H_2O_2$), hydroxyl radicals ($OH^.$), and singlet oxygen ($^1O_2$) are the ineluctable products of indispensable metabolic processes in aerobic organisms (*Schieber & Chandel, 2014*). In plants, ROS are generally from metabolic reactions including electron transport chains in chloroplasts and mitochondria, lipid catabolism in glyoxylic acid and peroxisomes, and photorespiration (*Asada, 2006*; *Dietz, Turkan & Krieger-Liszkay, 2016*). Current studies have been focusing on the toxicity of ROS for a long time as it can cause oxidative damage on DNA, RNA, protein and biomembranes (lipids) (*Hossain et al., 2015*). However, as a common molecule that exists in almost all aerobic organisms, ROS also carries great significance to support a cell function and plays significant roles in various major life activities such as development, proliferation, cell death and signaling (*Baniulis et al., 2013*). Among them, the study of signaling function of ROS is the most thorough and detailed.

Corresponding author
Yong Huang,
yonghuang@hunau.edu.cn

ROS usually works as a signal required for the cascade set off through stress sensor, and they can be produced by NADPH oxidase (also known as respiratory burst oxidase homologs, RBOHs), superoxide enzymes, and peroxidases from apoplasts (*Suzuki et al., 2011*).

To maintain the normal operation of plant function, the ROS content of each compartment in the plant needs to be kept accurate, which requires the balance between ROS production and clearance (*Choudhury et al., 2017*). Enzymatic antioxidant system plays an important role in ROS scavenging, but each subcellular compartment has its own ROS production and scavenging pathway, so the ROS homeostasis in each subcellular compartment is different, further hinting the unique characteristics of ROS (*Choudhury et al., 2017*). The major antioxidase system Ascorbate peroxidase (APX), Superoxide dismutase (SOD), Catalase (CAT) and Peroxidase (POD) and secondary metabolites Carotenoids involve in the ROS elimination (*Choudhury et al., 2017*; *Havaux, 2014*; *Wang, Lin & Al-Babili, 2021*).

Previously, the APXs, a group of important enzymes to balance ROS in cells have been identified and characterized in Arabidopsis. They are distributed in different subcellular compartments and serve distinct functions (*Heidari, 2010*; *Panchuk, Zentgraf & Volkov, 2005*). The APX in Arabidopsis chloroplasts includes sAPX (stromal APX) and tAPX (thylakoid APX) (*Jespersen et al., 1997*). The APX in chloroplasts have canonical ROS scavenging roles and is likely to simultaneously act as a ROS signaling regulator (*Maruta et al., 2016*). Microsomal APXs in Arabidopsis are APX3, APX4 and APX5, their functions are also mainly about ROS (especially $H_2O_2$) detoxification (*Panchuk, Zentgraf & Volkov, 2005*). As mentioned earlier, the source of ROS in the cytosol is the most diverse, and the signaling pathways are most complicated. Cytosol is also the main site of ROS-mediated stress response in plants, which makes function of cytosolic APX particularly important. Cytosolic APXs, APX1, APX2 and APX6 carry diverse functions (*Fryer et al., 2003*; *Karpinski et al., 1997*; *Kubo et al., 1995*; *Panchuk, Volkov & Schoffl, 2002*).

In the process that ascorbate peroxidase purges ROS, it has a higher affinity to $H_2O_2$ than its affinity to any other ROS, suggesting its main responsible role for $H_2O_2$ clearance (*Panchuk, Zentgraf & Volkov, 2005*). APX plays an essential role in ascorbate-glutathione cycle, the major hydrogen peroxide scavenging process in plants under stress (*Sofo et al., 2015*). Ascorbic acid (AsA) works as an electron donor when ascorbate peroxidase reduces hydrogen peroxide to $H_2O$, and is further oxidized to form monodehydroascorbate (MDHA). Part of MDHA will be reduced to AsA by monodehydroascorbate reductase (MDHAR) to be recycled for refunction; and another part of MDHA can be further oxidized to form dehydroascorbic acid (DHA). In addition to participating in the deoxidation of $H_2O_2$, APX is also involved in resistance to a variety of abiotic stresses. Overexpression of *Solanum melongena SmAPX* and *LcAPX* in Arabidopsis show greater flood resistance and less oxidative damage under a waterlogged condition (*Chiang et al., 2017*). Overexpression of *OsAPX* in *Medicago sativa L.* shows better salt tolerance (*Zhang et al., 2014*). Similarly, the survival rate of *Thellungiella salsuginea TsApx6* overexpressing Arabidopsis was increased and the leaf water loss rate was reduced under drought stress (*Li et al., 2016*).

With global climate change, plants often encounter relatively severe abiotic stresses, including drought stress, salt stress, low temperature stress, high temperature stress and toxic metals of aluminum, arsenic and cadmium in soil, which affect their growth and development, and even cause crop yield reduction and crop quality drop (*Zhu, 2016*). *Brassica napus* is an important economic crop and oil crop in China. The *B. napus* cultivar Xiang-you 15 (XY15) used in this experiment is bred by Hunan Agricultural University. It has the characteristics of low erucic acid and low glucosinolates and is cultured in the middle and lower reaches of Yangtze River of China mainly. In the resistance of plants to abiotic stress, APX plays a very important role, whereas the function of APXs in *B. napus* remains still unclear. Therefore, it is of great significance to study the role of *APXs* in *B. napus* for improving rapeseed varieties.

## MATERIALS AND METHODS

### Plant growth conditions and abiotic stress treatment

*Brassica naupus* XY15 were grown under the condition of 22 °C, 16 h light and 8 h dark photoperiod, 70% relative humidity. The 30-day seedlings (with 6 leaves) were treated with 300 mmol/L NaCl, 20% polyethylene glycol 6000 (PEG6000) and 40 °C heat for 3 h, 6 h, 9 h and 12 h, respectively. The leaf of three plants under each treatment, along with samples of no-treatment control group, were collected respectively. The samples were frozen in liquid nitrogen immediately and stored in −80 °C refrigerator to retain activity of RNA and other molecular.

### Screening and identification of *BnaAPX* genes

The amino acid sequences of *AtAPXs* in *Arabidopsis thaliana* were downloaded from the NCBI (http://www.ncbi.nlm.nih.gov/). The *AtAPXs* were used as queries to search for *BnaAPX* genes in *B. napus* Genome Browser of Genoscope (http://www.genoscope.cns.fr/brassicanapus/). The Expect Value was set at $1{\times}e^{-10}$ to the APX domain (IPR002016) by submitting them in InterProScan (http://www.ebi.ac.uk/interpro/) (*Tao et al., 2018*).

### Multiple alignment and evolutionary analysis of BnaAPXs

Multiple alignments of BnaAPX and AtAPX protein sequences were performed by ClustalX (2.0) (*Larkin et al., 2007*). Then, the file was subjected to MEGA7 software to construct phylogenetic tree with Neighbor-joining method. The tree construction setting was dependent on full length protein sequences of BnaAPXs and AtAPXs. Neighbor-joining method was set as follows: sites as pairwise deletion included; substitution model consisting of Poisson model; and Bootstrap test of 1,000 replicates for internal branch reliability (*Kumar, Stecher & Tamura, 2016*). 26 genes were identified as *BnaAPX* genes according to their homology with *AtAPX* genes. The final evolutionary tree was constructed with the multiple alignments of 26 *BnaAPX* genes, 8 *AtAPX* genes and 8 *OsAPX* genes. TBtools software (*Chen et al., 2020*) was used to exhibit the synteny relationships of *APX* genes in the *B. napus*, between *B. napus* and *B.rapa* or *B. oleracea*.

## Gene structure, chromosomal location and domain organization analysis of *BnaAPX* family

The cDNA and DNA sequences of *BnaAPX* family genes were obtained in *B. napus* Genome Browser of Genoscope. And the analysis of their exon-intron structures was accomplished by submitting them in the Gene Structure Display Server 2.0 (GSDS) website (http://gsds.cbi.pku.edu.cn/) (*Hu et al., 2015*).

The chromosomal location information of all the *BnaAPX* genes were obtained from *B. napus* Genome Browser and analyzed by MapChart (version 2.23) software (*Voorrips, 2002*).

The protein sequences of BnaAPXs were submitted to the Simple Modular Architecture Research Tool (SMART version 8, http://smart.embl.de/) for identification and annotation of the conserved domains (*Letunic & Bork, 2018*). PFAM database was chosen in SMART version 8 to obtain the protein sequence of domains.

## Responses of BnaAPX genes to abiotic stress

*B. napus* sample RNA was extracted by Trizol reagent (MagZol[TM] Reagent, Magen), and then DNA was cleaned by RNase-Free DNase I in RevertAid RT Kit (Thermo Scientific). RNA samples were reversed into cDNA by RiboLock RNase Inhibitor and RevertAid RT in RevertAid RT Kit. These cDNA worked as templates for quantitative real-time PCR (qPCR) to detect the expression of relative genes, the qPCR primers were designed by primer premier 5 (File S1). The qPCR reaction is performed with UltraSYBR Mixture (High ROX) (Cwbio), which contains GoldStar Taq DNA Polymerase, PCR Buffer, dNTPs, SYBR Green I fluorescent dyes, etc. Three-step quantitative real-time PCR program was performed as: 95 °C for 10 min, following 40 cycles of 95 °C for 10 s, 56–64 °C for 30 s and 72 °C for 32 s. Along with the melting curve procedure to detect the specificity of primer: 95 °C for 15 s, 60 °C for 1 min, 95 °C for 15 s and 60 °C for 15 s. Expression data acquired were normalized with the actin gene of *B. napus* and calculated with the $2^{-\Delta\Delta CT}$ method to analyze the relative changes in gene expression (*Livak & Schmittgen, 2001*).

## Measurement of the enzyme activity of APX

The activities of APX were determined according to the protocol of ascorbate peroxidase activity assay kit (Solarbio, Beijing, China). And the content of $H_2O_2$ were assayed using commercial kits provided by Beyotime (Beyotime, Shanghai, China). In brief, all the 30-day seedlings were divided into three groups: high salt group (300 mmol/L NaCl), water deficient group (20% polyethylene glycol) and high temperature group (40 °C heat). Three replicates were used in each group, and the leaves were collected every 3 h. The leaves (1.0 g) of three plants under each treatment were collected as samples, and were ground in liquid nitrogen with reagent I (1.0 ml) in ice bath. Subsequently, the extraction was centrifuged at 13,000 g at 4 °C for 20 min, and the supernatant was used for the enzyme activity measurement. The products were determined by BioTek's Synergy ek's Synergy HT Reader (BioTek Instruments, Inc., Highland Park, PO Box 998, Winooski, VT 05404-0998). Each activity unit was defined according to the instruction of the antioxidant enzyme assays kit. The enzyme activity of APX and content of $H_2O_2$ would replicate three times, and the data would be analyzed by Excel (*Caverzan et al., 2012*; *Shigeoka, Nakano & Kitaoka, 1980*).

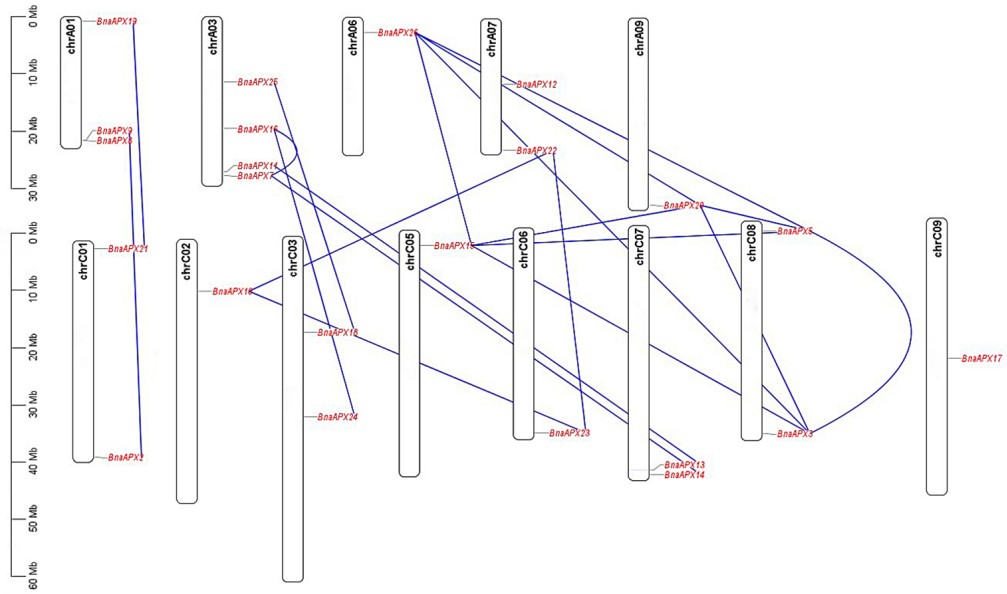

**Figure 1** **Chromosomal location of *APX* genes in *B. napus*.** Twenty-three of the 26 *APX* genes have been mapped on chromosomes A01-A09 and C01-C09. The chromosome map was constructed using the MapChart (2.23) software. Blue lines indicate duplicated *BnAPX* gene pairs. The scale on the chromosome represents megabases (Mb) and the chromosome number is indicated at the top of each chromosome.

## RESULTS

### Identification and chromosomal location of *BnaAPX* genes in *B. napus*

Twenty-six *BnaAPX* genes were identified on the basis of their homology with *AtAPX* genes and nominated as from *BnaAPX1 to BnaAPX26* based on their gene accession number (File S2). *BnaAPX* gene is widely distributed in 13 out of 19 chromosomes in *B. napus* (Fig. 1). Eleven *BnaAPXs* located in five chromosomes of A sub-genomes (come from *B. pekinensis*) and 12 *BnaAPXs* located in eight chromosomes of C sub-genomes (come from *B. oleracea*). Among them, ChrA03 carried four *BnaAPXs*, ChrA01 carried three *BnaAPXs*, and five chromosomes (chrA07, C01, C03, C07 and C08) carried two *BnaAPXs* each. In addition, six chromosomes (chrA06, A09, C02, C05, C06 and C09) carried one *BnaAPX*. No *BnaAPXs* were distributed on chromosomes A02, A04, A05, A08, A10 and C04.

The length of BnaAPX protein sequences ranged from 197 (BnaAPX1) to 439 (BnaAPX22 and BnaAPX23) amino acids, molecular weight (Mw) varied from 21.76 kDa (BnaAPX1) to 47.47 kDa (BnaAPX22), and isoelectric point (pI) ranged from 4.72 (BnaAPX1) to 8.73 (BnaAPX18), respectively (File S2). The intron-exon structure of the *BnaAPX* genes in each clade was described separately. It was obvious that not only the homologous genes in *B. napus* had a similar gene structure, but also the *AtAPX* genes (Fig. 2; File S3).

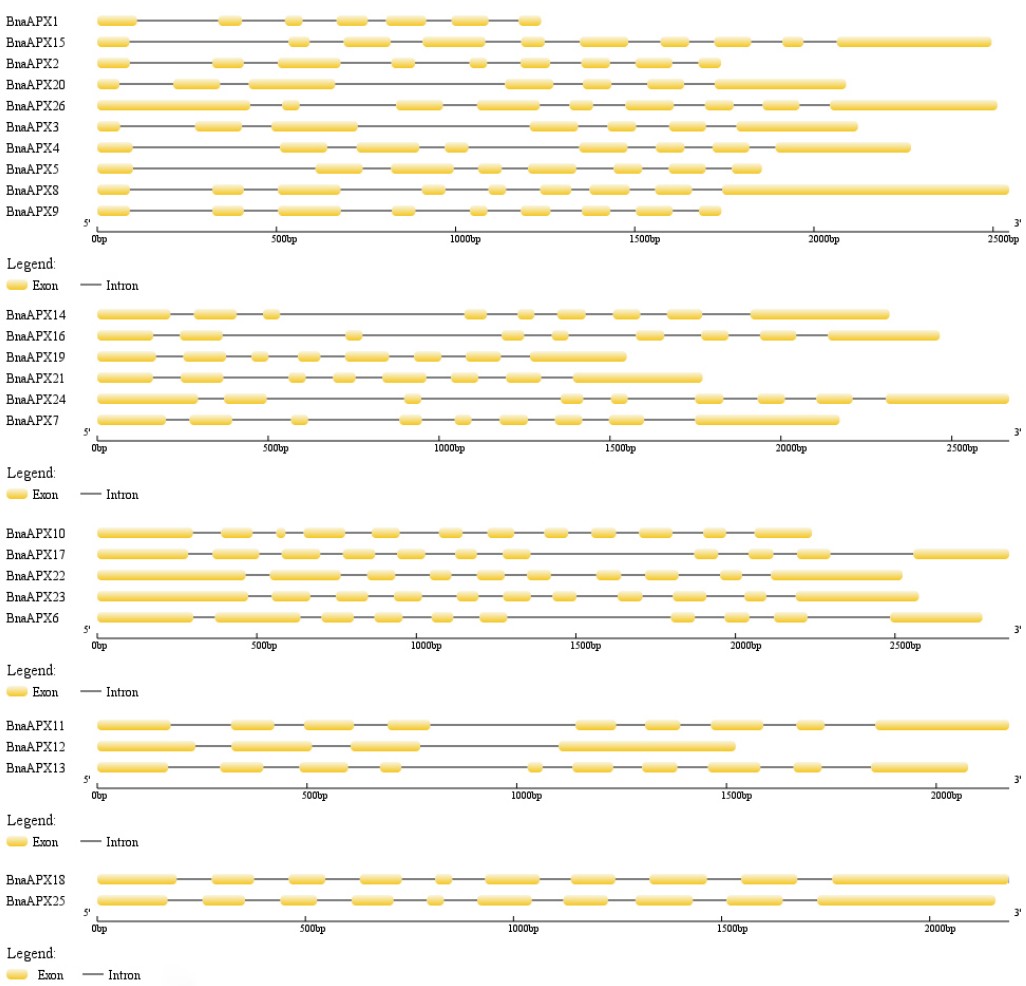

**Figure 2** **Gene structure of *APX* genes in *B. napus*.** The gene structure of 26 *APX* genes was constructed by Gene Structure Display Server 2.0 (http://gsds.cbi.pku.edu.cn/). Yellow boxes represented exons and black lines of the same length represented introns. The sizes of exons can be estimated by the scale at bottom.

## Phylogenetic relationship of *BnaAPX* genes

A neighbor-joining evolutionary tree of *BnaAPX* gene family was constructed with 26 amino acid sequences of *BnaAPX* s, eight amino acid sequences of *AtAPX* s and eight amino acid sequences of *OsAPX* s. The results showed that all *APXs* was grouped into five clades (Fig. 3), among which *BnaAPX1-5*, *8*, *9*, *15*, *20*, *26* and *AtAPX1*, *AtAPX2*, *OsAPX1*, *OsAPX2* were clustered into clade I, *BnaAPX11* and *BnaAPX13* were clustered into clade IV with *AtAPX6*, clade I and clade IV were located in cytoplasm coping with stress from sulfur dioxide, ozone, heat and high light (*Fryer et al., 2003*; *Karpinski et al., 1997*; *Kubo et al., 1995*; *Panchuk, Volkov & Schoffl, 2002*). *BnaAPX7*, *14*, *16*, *19*, *21*, *24* and *AtAPX3*, *AtAPX5*, *OsAPX3*, *OsAPX4* were clustered into clade II, *BnaAPX25* and *BnaAPX18* were clustered into clade V with *AtAPX4*, clade II and clade V were located in microsomes, role

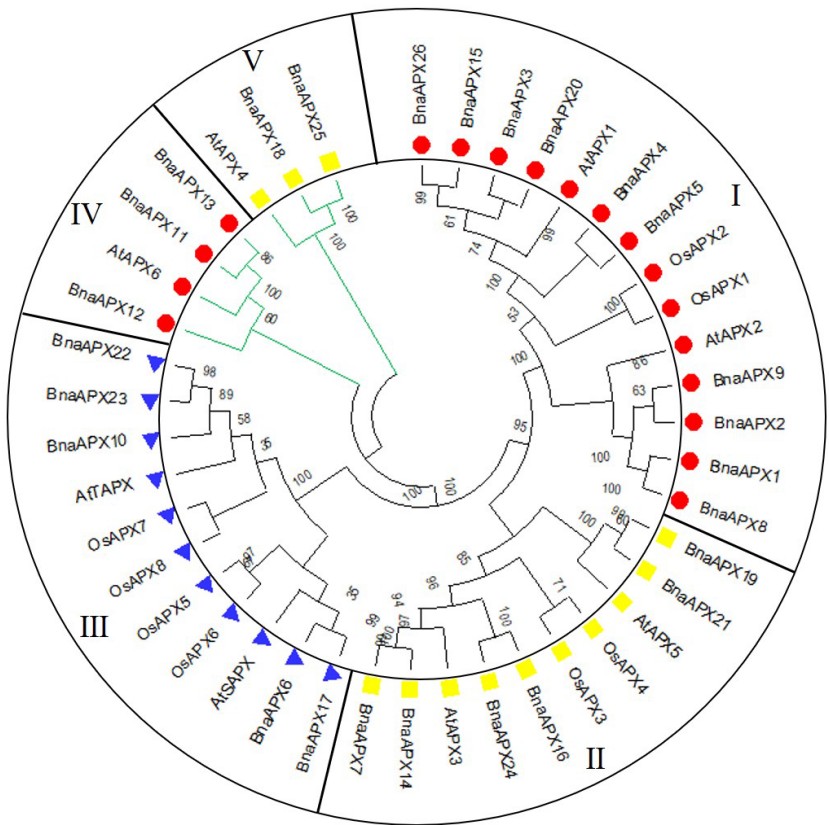

**Figure 3  Phylogenetic tree of APX proteins in *B. napus*, *A. thaliana* and *O. sativa*.** This tree includes eight APX proteins from *A. thaliana* (AtAPX), 8 APX proteins from *O. sativa* (OsAPX) and 26 from *B. napus* (BnaAPX). The tree was constructed using MEGA 7.0, and the bootstrap test replicate was set as 1,000. Five clades were named as sub-family I–V marked with different colours.

in ROS (especially $H_2O_2$) detoxification (*Panchuk, Zentgraf & Volkov, 2005*). *BnaAPX6, 17, 10, 22, 23* were clustered into clade III with *AtSAPX, AtTAPX, OsAPX5, OsAPX6,* and *OsAPX7, OsAPX8* were located in chloroplast as ROS signaling regulators and detoxidotes (*Maruta et al., 2016*).

## Conserved motif analysis of BnaAPX protein family

BnaAPX protein sequences are subject to MEME to identify the conserved motifs. 10 motifs have been found in BnaAPXs (Fig. 4). They are respectively named as motif 1–10. Among them, motif 1, 3–6, 9, 10 belong to plant_peroxidase_like superfamily, while motif 2 belongs to PLN02608 superfamily. Apparently, motif composition of BnaAPX protein in the same clade resembles each other. Clade I has almost motifs 1–9. Most members of clade II have the same composition and order of motifs with clade I. similarly, Clade III has almost all of 10 motifs except for motif 9, apart from that, motif 1, 4, 5, 7, 8, 10 are encoded in most members of clade IV, and motif 2, 6, 8 are shared in genes of clade V. All microsomal APX proteins contained the same composition and sequence of motif 2, motif 6 and motif 8. All chloroplast APX proteins contained the same composition and sequence

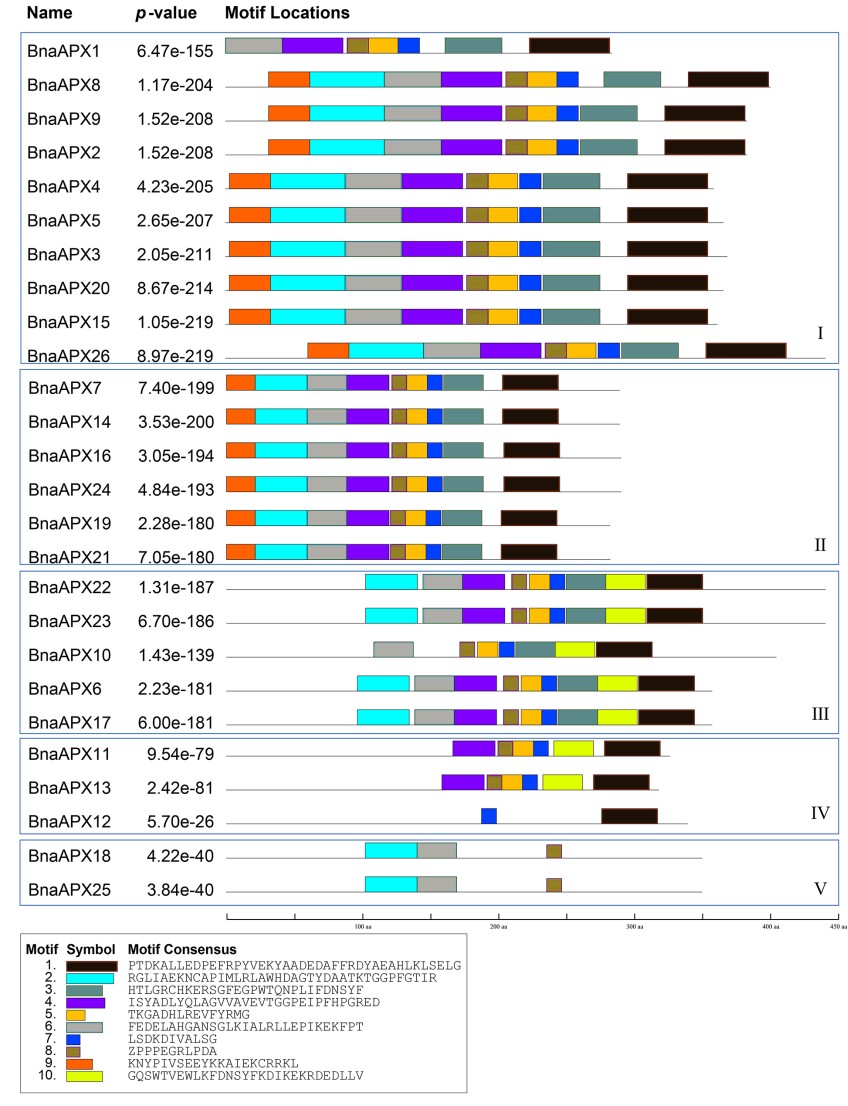

**Figure 4** **Conserved motifs of APX proteins in *B. napus*.** Ten predicted motifs were represented by different coloured boxes. The sequence information for each motif is provided in the bottom.

of motif 1, 3, 5–8,10. Almost all cytoplasmic APX proteins contained motif 1, 4, 5, 7, 8, except BnaAPX12 which contained only motif 7 and motif 1. Motif 2, 4, 6 , 8 exist in a majority of BnaAPX proteins, which indicates that they are conserved motifs of BnaAPXs. In addition, motif 2 contains a typical APX active site (APLMLPLAWHSA), and motif 7 contains a peroxidase proximal heme ligand domain (DIVALSGGGHTL). Meanwhile, motif 2 and motif 4 have relatively conserved structures.

## Gene expression of *BnaAPX* genes under abiotic stress

In order to study the expression pattern of *APX* gene in the leaves of *B. napus* seedlings under salt, high temperature and PEG treatment (to mimic osmotic stress), the expression profile of *BnaAPXs* in the samples was analyzed by qPCR (Fig. 5A). Microsomal *APX* s

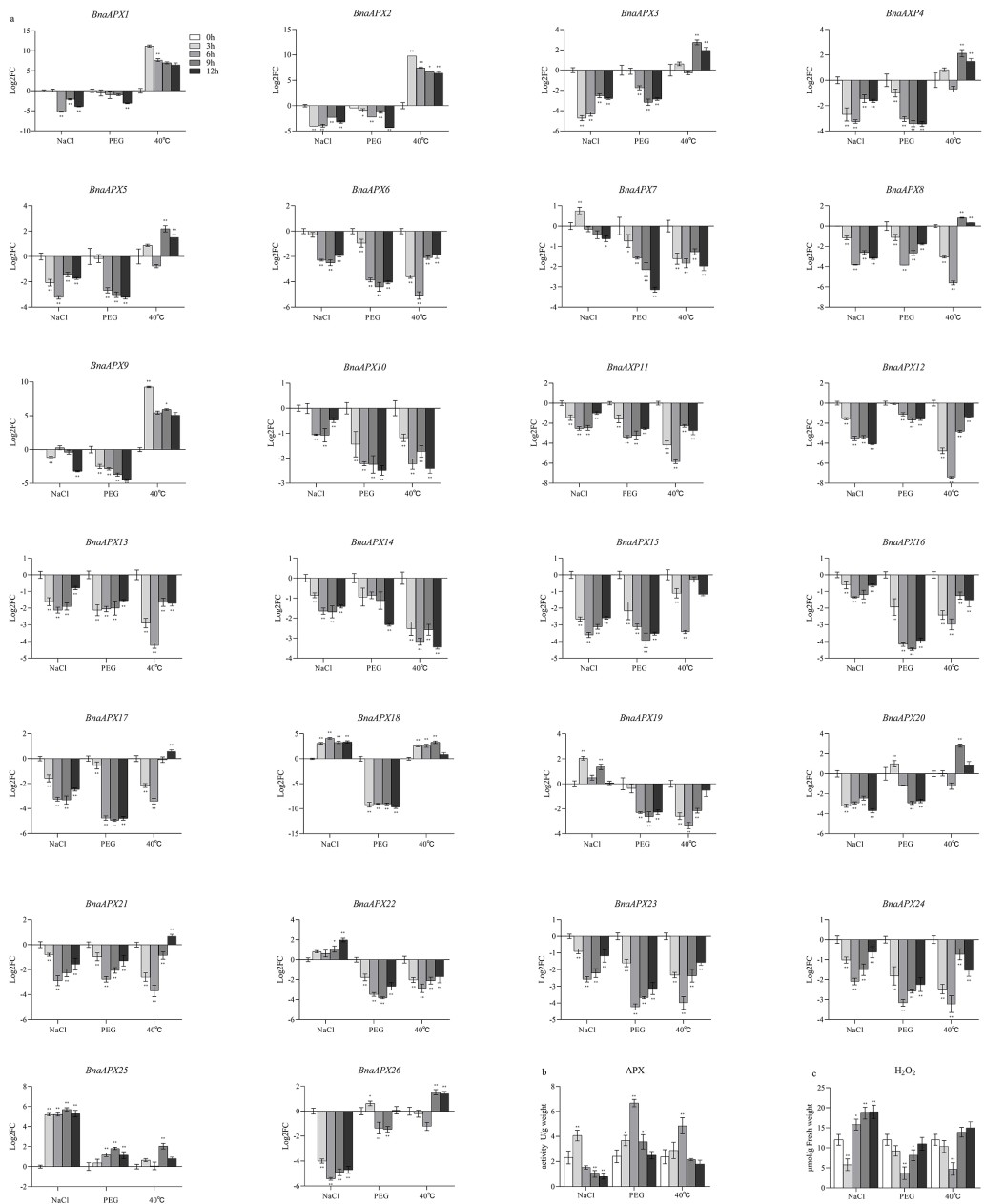

**Figure 5 The relative expression of APX genes, the activities of APX and the content of H2O2 under three treatments in *B. napus*.** (A) The relative expression of APX genes, the actin gene of B. napus was used as an internal reference; (B) The activities of APX; (C) The content of H2O2. All data were representative of three independent experiments, $N = 3$ for each group. Data are presented as the mean ±standard deviation (SD). An asterisk represent corresponding gene significantly up- or down-regulated by Student's $t$-test between the treatment and the control. *$P < 0.05$, **$P < 0.01$. ***$P < 0.001$.

and chloroplast *APX* s in *B. napus* leaves were almost not expressed under PEG treatment, except that *BnaAPX2* 5 was highly expressed in every group; Cytosolic *APX* s were barely expressed under PEG treatment except for *BnaAPX20* and *BnaAPX26* genes. In the NaCl-treated group, most of the microsomal *APX* s were expressed at a low level, except for the expression of Clade V and *BnaAPX19*; and the expression of chloroplast *APX* s was also rare, except for *BnaAPX22*, which was highly expressed; however, there were almost no expression of cytosolic *APX* s has been found in group. On the contrary, the case of the 40 °C high temperature treatment stress group was completely different. The expression of cytosolic *APX* s in *B. napus* leaves were in a high level in high temperature environment, except the genes of clade IV and *BnaAPX15*. Similarly, The expression level of microsomal *APX* s and chloroplast *APX* s were also generally low under high temperature, except that the gene expression level of clade V was relatively higher.

In this experiment, microsomal *APXs* were mainly involved in the response to salt stress, among which *BnaAPX18*, *BnaAPX 19* and *BnaAPX 25* were the most important. Cytosolic *APX* s was mainly involved in the response to high temperature stress, especially *BnaAPX 1-5, 9, 20*. In addition, only *BnaAPX 22* in chloroplast *APX* showed high expression under salt stress. And only *BnaAPX 25* in microsomal *APX* was highly expressed under PEG stress.

### Activity of APX under abiotic stress treatment

The activity of APX in *B. napus* increased first and then decreased under salt, PEG and high temperature treatment (Fig. 5B). The activity reached the maximum value after 3 h of salt treatment, but dropped to less than half of the initial value at 12 h. Under PEG treatment, the activity reached the maximum after 6 h, and the activity was still higher than the initial value at 12 h. Under high temperature stress, the activity reached the maximum at 6 h, and then slightly lower than the initial value at 12 h. In addition, the content of $H_2O_2$ decreased first and then increased which was basically opposite to APX activity (Fig. 5C).

## DISCUSSION

### The evolution and conservation of BnaAPXs

APX plays an important role in scavenging $H_2O_2$ and maintaining the balance of ROS content in plants, which is of great significance for the growth and development of life and resistance to external stresses. Previously, eight *APX* genes have been identified in *A. thaliana*, which serve different functions at different subcellular locations (*Panchuk, Zentgraf & Volkov, 2005*). Based on *AtAPX* gene reference, 26 *APX* homologous genes were found in *B. napus*. These genes are much richer than eight in *A. thaliana*, eight in *Oryza sativa* (*Teixeira et al., 2006*), nine in *Sorghum bicolor L* (*Akbudak et al., 2018*) and seven in *Solanum lycopersicum* (*Najami et al., 2008*). The number of *APX* genes in *B. napus* is the same as the number of *APX* genes in *Gossypium hirsutum* (*Tao et al., 2018*). Studies have shown that the genomes of *B. pekinensis* and *B. oleracea* are triploidized from *A. thaliana*, and habor three homologous genes of *A. thaliana* (*Wang et al., 2011*). *B. napus* (4n = 38) is crossed by *B. rapa* (2n =20) and *B. oleracea* (2n = 18) (*Morinaga, 1929*; *Morinaga, 1934*). In fact, *B. napus* carries much more *APXs* than its parents *B. rapa* (nine

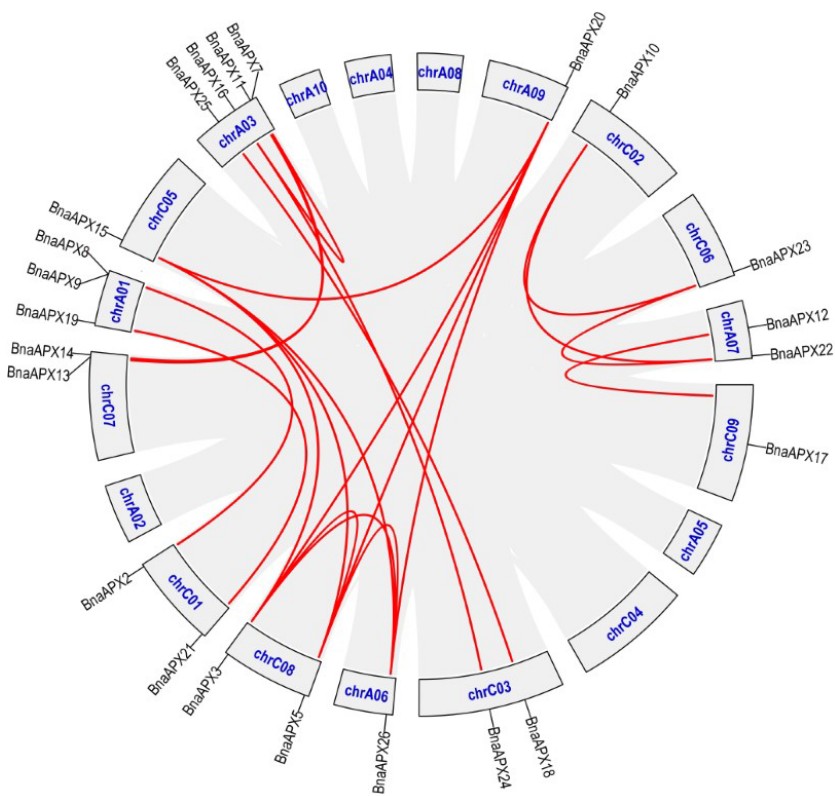

**Figure 6** **The synteny analysis of APX family in *B. napus*.** Gray lines indicate all synteny blocks in the *B.napus*. genome, and the red lines indicate duplicated *BnAPX* gene pairs. The chromosome number is indicated at the bottom of each chromsome.

*APXs*) and *B. oleracea* (10 *APXs*), but it is still less than the theoretical number of 38. *B. napus* inherited most of the parents *APX* genes. Some *APXs* lost during the formation of *B.napus*, while some *APXs* duplicated (*Udalll & Wendel, 2006*) (Fig. 6; File S4). For example, an *APX* gene on C07 chromosome of *B. oleracea* corresponded to only one gene on A07 chromosome of *B. napus*, and the homologous gene on C07 chromosome was missing. While a gene on chromosome A05 of *B. rapa* not only corresponded to a gene on chromosome A06 of *B. napus*, but also corresponded to genes on A09, C05 and C08.

Twenty-six *BnaAPX* genes formed five branches with homologous genes in *A.thaliana* and *O.sativa* (Fig. 3). By analyzing the structure of intron and exon, it was found that *BnaAPX* have a similar gene structure with *AtAPX* in each subfamilies (Fig. 2; File S3). Domain analysis showed that subgroup IV and subgroup V had similar structure in domain (Fig. 4), indicating that these sequences might have common ancestors. The difference between the two subfamilies indicates that the two subfamilies have relatively independent evolutionary history.

## The roles of BnaAPXs in abiotic stresses

The expansion of APX family in *B.napus* may be a result in adaptation to environment. Arabidposis APXs vary in subcellulcalization, including chloroplast, microsome and

cytosol (*Kubo et al., 1995*; *Maruta et al., 2016*; *Panchuk, Volkov & Schofl, 2002*). Based on the phylogenetic tree, it was inferred that clade I and clade IV function as cytosolic APX. clade function as chloroplasts APX; and Clade II and clade V function as Microsomal APXs (Fig. 3).

APX genes involved abiotic stresses by ROS scavenging or in ROS signal regulation (*Maruta et al., 2016*). Under high salt, water deficiency and high temperature stress, APX activity and $H_2O_2$ content show the opposite levels (Figs. 5B, 5C). At an early stage, APX was activited and enable to clear $H_2O_2$, but its antioxidant capacity gradually decreases later. At the same stage, qRT-PCR also showed that five *BnaAPX* genes were up-regulated rapidly and violently. Among them, *BnaAPX1*, *BnaAPX2* and *BnaAPX9*, homologs of *AtAPX1* and *AtAPX2*, maintains a high expression level under high temperature stress (Fig. 5A). It was speculated that these three genes could effectively remove oxides and improve the heat resistance of plants under high temperature stress. *BnaAPX18* and *BnaAPX25*, homologous to *AtAPX4* located in microsomes (*Panchuk, Zentgraf & Volkov, 2005*), showed high expression levels under high salt stress, suggesting that the genes had a certain role in scavenging peroxides such as hydrogen peroxide and improving salt tolerance of plants. *BnaAPX25* was significantly up-regulated under three abiotic stresses. It was predicted that *BnaAPX25* can regulate chloroplast photosynthetic system and scavenge reactive oxygen species under abiotic stress, and it was speculated that *BnaAPX25* can quickly respond to abiotic stress (*Panchuk, Zentgraf & Volkov, 2005*).

However, the expression levels of some genes showed a downward trend under stress. For example, the expression of *AtAPX2* was down-regulated under drought stress simulated by mannitol (*Li et al., 2019*). Similarly, the expression levels of most *BnaAPXs* changed little or even decreased under drought stress. It was speculated that the expression of these genes was inhibited after drought stress, and ROS was eliminated by other genes or other ways. In addition, the expression patterns of some genes were different under different stresses. For example, the expression levels of *BnaAPX1* and *BnaAPX2* were significantly up-regulated under high temperature stress and down-regulated under high salt stress. APX genes expression level little matched the APX activity or $H_2O_2$ content at the stress treatment process (Fig. 5B, 5C), suggesting that the APX actitivity might be governed by post-translational regulattion and the exact role of APX genes should be explored in depth.

## CONCLUSIONS

In conclusion, 26 *APX* genes were identified in *B. napus*, and their different biochemical characteristics were analyzed. The similar gene structure and motif arrangement of BnaAPX protein in these subfamilies further supported the classification predicted by phylogenetic tree. qRT-PCR analysis showed that *BnaAPX* gene could respond to a variety of abiotic stresses such as high salt, water shortage and high temperature stress at transcriptional level. The results can provide basic and comprehensive information for further functional analysis of *APX* gene family in *B. napus*.

### Funding
This work was funded by National Nature Science Foundation of China (31971834), Natural Science Foundation of Hunan Province (2019JJ40116). The funders had no role in study design, data collection and analysis, decision to publish, or preparation of the manuscript.

### Grant Disclosures
The following grant information was disclosed by the authors:
National Nature Science Foundation of China: 31971834.
Natural Science Foundation of Hunan Province: 2019JJ40116.

### Competing Interests
The authors declare there are no competing interests.

### Author Contributions
- Jiao Pan conceived and designed the experiments, performed the experiments, analyzed the data, prepared figures and/or tables, authored or reviewed drafts of the paper, and approved the final draft.
- Lei Zhang and Min Chen performed the experiments, analyzed the data, prepared figures and/or tables, authored or reviewed drafts of the paper, and approved the final draft.
- Yuxuan Ruan, Peifang Li, Zhihui Guo, Boyu Liu, Ying Ruan and Mu Xiao performed the experiments, prepared figures and/or tables, authored or reviewed drafts of the paper, and approved the final draft.
- Yong Huang conceived and designed the experiments, authored or reviewed drafts of the paper, and approved the final draft.

### Data Availability
The raw data is available in the Supplementary File.

### Supplemental Information
Supplemental information for this article can be found online at http://dx.doi.org/10.7717/peerj.13166#supplemental-information.

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
