# Peer review of "Identification and charactering of APX genes provide new insights in abiotic stresses response in Brassica napus"

_PeerJ, doi:10.7717/peerj.13166_

## Round 0.1 · original submission · Major Revisions

It is my opinion as the Academic Editor for your article - Part cytosol and microsomal APX genes of Brassica napus are involved in abiotic stresses response - that it requires a number of Major Revisions.

·

Basic reporting

APX is important for the growth and development of organism to resistant the external stress. In this paper, authors detailedly described the structures and phylogenetic relationship of APX in B.napus, and their functions in abiotic stress were also briefly studied. In general, the paper supplies a professional guidance for the further study and application of APX in B.napus. However, the discussion part is a little bit wordy and did not catch the key point of the discover in this paper, which is encouraged to discuss it more streamlined and in-depth. And also, the last paragraph of introduction should be described more logically. Fig. 6 should be reorganized, as the APX activities and the content of H2O2 are totally different with qPCR graph, they should be set up as independent sub-figure, i.e., Fig 6b and Fig6c. Figure legends also need to be revised to make figures understood easily.

Experimental design

The title is suggested to be modified, as the paper is mainly described the structures and phylogenetic relationship of APX, and the response of APX to abiotic stress only occupies small portion.

Validity of the findings

No comment.

Additional comments

Line 16: Brassica napus (B.napus), abbreviation statement for the first time.
Line 81-83: Suggest to move this part to the behind of ....crop quality (Zhu, 2016), and modify the words to make it more logical.
Line 222: Suggest to be replaced by " Activity of APX under abiotic stress treatment".
Line 229: Replace with "which was basically opposite to APX activity" . Go through all the paper to check the logical of sentences.
In discussion, suggest to speculate why the activity of APX in B.napus was increased firstly, and then decreased with diverse treatments,as showed in Fig 6.
Line 275: ...the gene has..., line 288, played...., etc, check all the manuscript to make the words more correct.
Go through the reference to make them unified.
Line 436-439: describe the mean of lines connection from one gene to another.
Line 452: remove " of the same length" as the length of introns is different. and the gene names on figure should be italic if possible to change.
Line 454-455: Describe the indication of black line in Figure 5.
Line 457: replace three treatments with various treatments, and reorganized Figure 6, i.e., enzyme activity was set as Fig. 6b, and H2O2 content was Fig. 6c.

·

Basic reporting

Major:
1. Line 35-48, the authors introduced the ROS background information; however, there is lacking the important information about carotenoids as they play critical roles in the chloroplasts in quenching ROS. Few references mat be considered: N. Nisar, et al. Carotenoid metabolism in plants. Mol. Plant, 8 (1) (2015), pp. 68-82; Havaux, M. Carotenoid oxidation products as stress signals in plants. Plant J. 79, 597–606 (2014); Wang, J. Y., Lin, P.-Y., & Al-Babili, S. On the biosynthesis and evolution of apocarotenoid plant growth regulators. Semin. Cell. Dev. Biol. 109, 3–11 (2020).
2. Line 164-173, is the localization of the un-mapping APX could be isoforms of some of the APX and that’s why some of the APXs could not be confirmed? In line 173 the authors claimed AtAPX has the same feature as BnaAPX, but there is no such a figure of AtAPX to support it. Also, figure 4 should be named figure 2.
3. There is lacking references from lines 175-185 where the authors claimed the functions of APX among different clades and do not repeat the same sentence in the discussion part as it is wordy.
4. Line 188, figure 5 should be figure 4 in the text. Please re-arrange the figures.
5. Line 203-229, if I understood well, there is no localization data as well as whole leaves for qPCR; if this is the case, the authors should not claim the cytosolic or microsomal APXs. Otherwise, they should provide such a dataset. Moreover, there is no statistical test in the transcript and biochemical results, which makes it difficult to assess the data. In addition, please mark those figures clearly linked to the text.
6. Line 251-254, the phylogenetic analysis may not relate to the evolutionary cluster but functions; please check this claim carefully.
7. There is no discussion about the biochemical results?

Minor:
1. I would recommend using the full name with an abbreviation, such as Ascorbate peroxidase (APX), instead of APX (Ascorbate peroxidase) in the text.
2. Strongly suggest the authors re-check and revise the grammar carefully and better to read by naïve speakers.
3. Line 67-68, there is a missing or typo: it is a higher “anity“
4. Gene should be italic and could delete gene, such as APX in line 88 rather than APX gene.

Experimental design

no comment

Validity of the findings

no comment'

---

## Round 0.2 · Minor Revisions

Dear Dr. Pan,

It is my opinion as the Academic Editor for your article - Identification and charactering of APX genes provide new insights in abiotic stresses response in Brassica napus - that it requires a number of Minor Revisions. Please address these changes and resubmit.

·

Basic reporting

Authors have revised all the manuscript based on the reviewers suggestions.

Experimental design

No comment

Validity of the findings

No comment

Additional comments

No comment.

·

Basic reporting

The authors have satisfactorily addressed all my and the other reviewers' comments , and I appreciated the authors for improving the text. I have no further major comments.

Experimental design

A minor point, please add the biological replicate numbers (e.g. n=3) in the legends as well as the detailed statistical description, such as Statistical analysis was performed using two-tail student t-test. Different letters denote significant differences (*p < 0.05, **p< 0.01, ***p< 0.001).

Validity of the findings

N/A

---

## Round 0.3 · Minor Revisions

Dear Dr. Pan,

Thank you for your submission to PeerJ.
I am writing to inform you that your manuscript - Identification and characterizing of APX genes provide new insights in abiotic stresses response in Brassica napus - is almost ready to be Accepted for publication with only a minor revision.

There are many places that need improvements of English expression and I only have a few suggestions (see the attached PDF file).

---

## Round 0.4 · accepted · Accept

This product is acceptable for a publication for PeerJ. Congratulation!